# Heritability of Teat Condition in Italian Holstein Friesian and Its Relationship with Milk Production and Somatic Cell Score

**DOI:** 10.3390/ani10122271

**Published:** 2020-12-02

**Authors:** Francesco Tiezzi, Antonio Marco Maisano, Stefania Chessa, Mario Luini, Stefano Biffani

**Affiliations:** 1Department of Animal Science, North Carolina State University, Raleigh, NC 27695, USA; f_tiezzi@ncsu.edu; 2Istituto Zooprofilattico Sperimentale Lombardia Emilia Romagna “Bruno Ubertini”—I.Z.S.L.E.R. Territorial Section of Lodi and Brescia Sector Diagnostic, Animal Health and Welfare, 26900 Lodi, Italy; antoniomarco.maisano@izsler.it (A.M.M.); mariovittorio.luini@ibba.cnr.it (M.L.); 3Dipartimento di Scienze Veterinarie, Università degli Studi di Torino, 10095 Grugliasco, Italy; stefania.chessa@unito.it; 4Consiglio Nazionale delle Ricerche (CNR), Istituto di biologia e biotecnologia agraria (IBBA), Via Edoardo Bassini, 20133 Milano, Italy

**Keywords:** dairy cattle, teat-end hyperkeratosis, udder health, somatic cell, genetic correlation, selection response

## Abstract

**Simple Summary:**

Mammary infections in dairy cattle are still a major problem which impair animal health and jeopardize breeders’ efforts to attain sustainable production. The first natural protection against pathogens ‘access to udder tissue is the teat canal. Teat canal morphology can be influenced by several environmental factors but the present study confirmed the existence of a genetic component. Moreover it was observed that the teat canal morphology is related to milk production and somatic cell count, the latter being an indirect indicator of mammary infections. Considering that the current selection objectives implemented in dairy cattle worldwide have shifted toward a more balanced breeding goal, with a larger emphasis on health traits, a further genetic deterioration in teat condition is not expected.

**Abstract:**

In spite of the impressive advancements observed on both management and genetic factors, udder health still represents one of most demanding objectives to be attained in the dairy cattle industry. Udder morphology and especially teat condition might represent the first physical barrier to pathogens’ access. The objectives of this study were to investigate the genetic component of teat condition and to elucidate its relationship with both milk yield and somatic cell scores in dairy cattle. Moreover, the effect of selection for both milk yield and somatic cell scores on teat condition was also investigated. A multivariate analysis was conducted on 10,776 teat score records and 30,160 production records from 2469 Italian Holstein cows. Three teat scoring traits were defined and included in the analysis. Heritability estimates for the teat score traits were moderate to low, ranging from 0.084 to 0.238. When teat score was based on a four-classes ordinal scoring, its genetic correlation with milk yields and somatic cell score were 0.862 and 0.439, respectively. The scale used to classify teat-end score has an impact on the magnitude of the estimates. Genetic correlations suggest that selection for milk yield could deteriorate teat health, unless more emphasis is given to somatic cell scores. Considering that both at national and international level, the current selection objectives are giving more emphasis to health traits, a further genetic deterioration in teat condition is not expected.

## 1. Introduction

Udder health still represents one of the most demanding objectives to be attained in the dairy cattle sector. In spite of the impressive advancements observed on both management and genetic factors, the diseases of the mammary gland in response to an infection, i.e., a clinical (CM) or a subclinical mastitis (SCM), is one of the most common and expensive diseases faced [1,2,3,4,5,6]. In 2015 the estimated overall cost per case of clinical mastitis in the first 30 days of lactation was $444 [7]. Moreover, 71% of the costs of a case of clinical mastitis were indirect costs. Among the several factors which can mitigate or increase the occurrence of intramammary infections, udder and teat morphology play a substantial role [8,9,10,11]. Indeed, the most relevant pathogens affecting dairy cattle, e.g., *Staphylococcus aureus (S. aureus)* [12] gain access to udder tissues primarily via the teat canal. Teat-end hyperkeratosis (THK), i.e., the hyperplasia of the teat orifice’s keratin layer [13], is considered one of the most common teat condition changes observed in dairy cattle [14]. THK is commonly evaluated using an ordinal scale ranging from 1 (no THK) to 4 (severe THK) [15]. Results presented in a recent review show that a condition of severe THK is an important risk factor for both CM and SCM, while a mild THK can mitigate possible SCM [11]. Alterations in teat condition across lactation and parities are expected as a consequence of machine milking [14] but the existence of a genetic component has also been investigated [16,17]. In a first study enrolling 1740 US Holstein cows from nine herds, Chrystal and colleagues [16] found that heritability estimates of teat end shape for first, second, and all lactations combined were 0.53, 0.44, and 0.56, respectively. Teat diameter heritability estimates were 0.23, 0.27, and 0.35, respectively across the three lactations. Teat-end shape and diameter repeatability estimates were 0.75 and 0.36, respectively. In a subsequent study with 1259 cows from a single herd, Chrystal et al [17] found heritability estimates of teat-end shape for first, second, and third and later lactations of 0.34, 0.21, and 0.13, respectively. Repeatability ranged from 0.34 to 0.46. 

An additional and widely used phenotypic indicator of udder health is somatic cell count (SCC). After being transformed to somatic cell score (SCS) using a logarithmic transformation [18], it is used as a predictor in the selection for mastitis resistance [3,19]. Some studies [20,21,22,23,24,25,26,27] have investigated the relationship between THK and SCC and most of them reported positive associations, especially when THK is scored as severe. However THK scoring can be confounded with other factors, e.g., parity, and its relationship with SCC might not be clearly identified. Chrystal and colleagues investigated the relationship between teat end shape and SCS in two subsequent studies [16,17]. They used SCS as a dependent variable in a mixed model and included teat end shape as a covariate. In both cases, the effect of teat end shape was not significant. Among the possible explanations the authors hypothesized some genetic factors that would control SCS and prevail over the effect of teat end shape. 

The main objectives of this work were to further investigate the genetic component of THK and to elucidate its relationship with both milk yield (MY) and SCS. Moreover, the genetic change in THK when selecting for both MY and SCS was also investigated.

## 2. Materials and Methods

### 2.1. Dataset

Animal welfare and use committee approval was not needed for this study as datasets were obtained from pre-existing databases based on routine animal recording procedures. Data used in the present study were collected between September 2011 and August 2012 and between March 2016 and January 2017 in 48 Italian Holstein dairy farms located in the Lombardy region (Northern Italy, 45.4791° N, 9.8452° E). The average herd size was 106 milking cows, ranging from 38 to 285 cows. The original data set included 3087 cows [28].

Teat condition score (TCS) was evaluated visually for each cow during milk sampling and assigned a score using the methodology proposed by Neijenhuis et al. [15]: an absent callosity was evaluated as TCS = 1, a smooth callous ring around the orifice was evaluated as TCS = 2; a rough and very rough callous rings were evaluated as TCS = 3 and 4, respectively. Scores were applied on each single teat, generating four records for each cow scored, each one was assigned to the respective teat position (FL: front left; FR: front right; RL: rear left; RR: rear right). In addition to scoring values, date of scoring and hygiene scoring were obtained. Hygiene of udder, flanks and legs was scored based on a 4-point scale system [29], from very clean (score 1) to very dirty skin (score 4). Both TCS and hygiene score were collected by a group composed of four technicians who were previously trained to harmonize the scoring procedure.

Production records for daily milk yield (MY) and SCC per ml were obtained by the Regional Breeders’ Association (ARAL, Crema, Italy). The variable SCC was log-transformed into SCS by using the formula SCS = log_2_(SCC/100,000) + 3 [18]. Together with production records, calving dates, herd number and pedigree information were obtained. Using calving date, it was possible to extract the parity order and days in milk at teat scoring and production recording. 

Teat scoring was organized in a dataset where the four quarters appeared as repeated information of the same phenotype. Each herd was visited and scored only once although the scoring was conducted over two consecutive days in two of the herds. Teat position information was assigned to each record, so that each cow showed four records with the respective position indicator. Only cows showing all four records were included in the analysis. 

Production records from the same cows were retained and were organized as repeated measures on the same individuals. For each cow, only production records from the same lactation number of the scoring were retained. The median number of records per cow was 10, each record reported MY and SCS for that cow in that test-day.

The final dataset was built stacking the teat score and production datasets. Records that showed teat scoring did not show production records, and vice versa.

The final dataset included 10,776 scoring records and 30,160 production records, for a total of 40,936 records available for analyses. All records came from 2649 cows, daughters of 869 sires and 2408 dams, while 275 cows appeared also as dams. All animals were raised in a total of 48 herds.

The choice of building a dataset where test scores and production records were not aligned was dictated by practical modeling decisions. The alignment of the teat score records with the closest production record could have been done. However, the repeated nature of the teat score record would have forced the production record to be repeated four times, which was not appropriate. At the same time, repeated records for both teat scores and production records allowed us to disentangle the additive genetic and permanent environmental effects. By not aligning records, the residual covariance between the teat scores and the production records could not be estimated. Nevertheless, such covariance should be forced to zero, because the phenotypes came from different data capture and recording systems. The cow permanent environmental effect allowed us to estimate a non-genetic covariance between the two sets of traits.

### 2.2. Statistical Analysis

Three teat scoring traits were defined and included in the analysis. A four-classes ordinal scoring (1, 2, 3, 4) was maintained for the trait hereinafter called TS. In addition, two binary traits were created. Classes 2, 3 and 4 were combined into a single class creating the trait hereinafter called TS_a_; classes 1 and 2 vs. 3 and 4 were combined to create trait TS_b_. Relative frequencies for all trait classes are reported in Table 1. 

A tri-variate threshold-linear model was used for analyzing traits combination such as TS-MY-SCS, TS_a_-MY-SCS, TS_b_-MY-SCS. Model specifications were:$$y\left(\lambda \right)=Xb+Z_{h}h+Z_{p}p+Z_{s}s+e$$
where y is the phenotype vector for MY and SCS, λ is the underlying liability for TS, TS_a_ or TS_b_, X and b are the incidence matrix and vector of solutions for fixed effects, $Z_{h}$ and h are the incidence matrix and vector of solutions for the herd environmental random effect, $Z_{p}$ and p are the incidence matrix and vector of solutions for the cow permanent environmental random effect, $Z_{s}$ and s are the incidence matrix and vector of solutions for the sire additive genetic random effect, e is the vector of residuals. Fixed effects common to all traits were parity by stage of lactation class (36 classes), defined by pasting the parity class (first, second and later lactations) with the stage of lactation month (last class including records until the 15th month of lactation). Additional fixed effects for TS, TS_a_ and TS_b_ were hygiene (4 classes) and teat position (4 classes). The vector of solutions for random effects were assumed to be normally and independently distributed. Specifically for the herd random effect:$$\left[\begin{array}{c} h_{1}\\ h_{2}\\ h_{3} \end{array}\right]\sim N\left(0,\;I⊗H\right)$$
where I is an identity matrix and H is the following variance-covariance matrix:$$\left[\begin{array}{ccc} \sigma_{h11}^{2} & \sigma_{h12} & \sigma_{h13}\\ \sigma_{h12} & \sigma_{h22}^{2} & \sigma_{h23}\\ \sigma_{h13} & \sigma_{h23} & \sigma_{h33}^{2} \end{array}\right]$$

Similarly, for the cow permanent environmental effect: $$\left[\begin{array}{c} p_{1}\\ p_{2}\\ p_{3} \end{array}\right]\sim N\left(0,\;I⊗P\right)$$
where I is an identity matrix and P is the following variance-covariance matrix:$$\left[\begin{array}{ccc} \sigma_{p11}^{2} & \sigma_{p12} & \sigma_{p13}\\ \sigma_{p12} & \sigma_{p22}^{2} & \sigma_{p23}\\ \sigma_{p13} & \sigma_{p23} & \sigma_{p33}^{2} \end{array}\right]$$

For the sire additive genetic effect: $$\left[\begin{array}{c} s_{1}\\ s_{2}\\ s_{3} \end{array}\right]\sim N\left(0,\;A⊗S\right)$$
where A is the numerator relationship matrix constructed on the pedigree and S is the following variance-covariance matrix:$$\left[\begin{array}{ccc} \sigma_{s11}^{2} & \sigma_{s12} & \sigma_{s13}\\ \sigma_{s12} & \sigma_{s22}^{2} & \sigma_{s23}\\ \sigma_{s13} & \sigma_{s23} & \sigma_{s33}^{2} \end{array}\right]$$
and for the residual effect: $$\left[\begin{array}{c} e_{1}\\ e_{2}\\ e_{3} \end{array}\right]\sim N\left(0,\;I⊗R\right)$$
where I is an identity matrix and R is the following variance-covariance matrix:$$\left[\begin{array}{ccc} 1 & 0 & 0\\ 0 & \sigma_{e22}^{2} & \sigma_{e23}\\ 0 & \sigma_{e23} & \sigma_{e33}^{2} \end{array}\right]$$

Therefore, residual variance for any teat scoring trait was fixed to 1 for identifiability in the threshold liability model and the covariance between the teat scoring and production traits was fixed to 0, due to lack of records showing both traits.

Variance components estimates were obtained using software THRGIBBS1F90 version 3.1 [30]. Solutions for TS_a_ and TS_b_ were then obtained using the package ‘MCMCglmm’ implemented in R version 3.6.1 [31] by fixing the variance components to the estimated values. This package was chosen for easier manipulation of solutions in the R environment. In both cases, a total of 300,000 iterations were run, removing the first 50,000 as burn-in and thinning every 50 iterations. Convergence was assessed by visual inspection of trace plots and Geweke’s test conducted using the ‘coda’ package in R [32]. 

### 2.3. Genetic Parameters

Heritability was calculated as: $$h^{2}=\frac{4\sigma_{s}^{2}}{\sigma_{s}^{2}+\sigma_{p}^{2}+\sigma_{h}^{2}+\sigma_{e}^{2}}$$

Intra-herd heritability was calculated as: $$h_{IH}^{2}=\frac{4\sigma_{s}^{2}}{\sigma_{s}^{2}+\sigma_{p}^{2}+\sigma_{e}^{2}}$$

Cow repeatability was calculated not considering the genetic effect as a source of cow repeatability, therefore: $$r^{2}=\frac{\sigma_{p}^{2}}{\sigma_{s}^{2}+\sigma_{p}^{2}+\sigma_{h}^{2}+\sigma_{e}^{2}}$$

Herd repeatability was calculated as: $$he^{2}=\frac{\sigma_{h}^{2}}{\sigma_{s}^{2}+\sigma_{p}^{2}+\sigma_{h}^{2}+\sigma_{e}^{2}}$$

In all formulas, $\sigma_{e}^{2}$ was ‘1’ for the teat score traits. Genetics, cow and herd correlations were calculated by dividing the covariance by the product of the standard deviations. Parameters were calculated at every iteration and posterior mean and 95% empirical confidence intervals were used as estimates of their standard error.

Least square means for all effects were calculated on the liability scale at every iteration and transformed to the phenotypic (probability) scale. Posterior mean and 95% empirical confidence intervals were used as estimates and their error. Plots were created using package ‘ggplot2’ implemented in R [33]. 

### 2.4. Genetic Change in Teat Score Traits

Expected selection response on the teat score traits was calculated for economic indices that combined MY and SCS with different values of relative emphasis. One economic index for each of the three teat score traits was built, including three traits (the teat score traits, MY and SCS). Relative emphasis was applied on MY and SCS, starting with 100% emphasis on MY and 0% emphasis on SCS and ending with 0% emphasis on MY and 100% on SCS, with step increases of 1% unit. While MY was given positive emphasis, SCS was given negative emphasis. Following [34,35,36] the genetic response on the teat score trait was calculated as:$$r=\frac{{w^{\prime }}_{g{}^{\circ}1}}{\sqrt{w^{\prime }Gw}}$$
where r is the genetic response for the teat score trait (TS, TS_a_ or TS_b_); w is the vector of index weight; G is the variance-covariance between the teat score trait, MY and SCS; $g_{{}^{\circ}1}$ is the first column of G. The genetic response was then expressed in genetic standard deviation units for the respective teat score trait.

## 3. Results

### 3.1. Data

Descriptive statistics for the teat score traits are in Table 1. Mean values and SD for daily milk production (MY) and daily linear score (SCS) were 29.73 ± 9.05 for and 3.97 ± 1.99 (not reported in table). The observed phenotypic trend for MY and SCS by parity and month of lactation are in Figure 1 and Figure 2, respectively.

### 3.2. Heritability and Environmental Effects Estimates

Estimates of variance components, heritability, intra-herd heritability, cow repeatability and herd repeatability for all traits are reported in Table 2. Estimates for MY and SCS came from the model where TS was the correlated trait, though posterior distribution estimates for production traits in the other models largely overlapped the values shown.

Heritability estimates for the teat score traits were moderate to low, with TS showing the largest value (0.238) followed by TS_a_ with 0.162 and TS_b_ with 0.084. A similar but weaker trend was found for the intra-herd heritability, TS showed an estimate of 0.289, TS_a_ showed 0.248 and TS_b_ showed 0.250. The weaker trend was due to the herd effect also decreasing from 0.221 for TS to 0.159 for TS_a_ to 0.093 for TS_b_. Cow repeatability (free of genetic variance) was 0.554 for TS, 0.46 for TS_a_ and 0.251 for TS_b_.

Heritability and intra-herd heritability estimates were larger for SCS than MY (0.146 vs. 0.065 and 0.234 vs. 0.126, respectively). Also cow repeatability was stronger for SCS than MY (0.368 vs. 0.257) while herd effect was weaker for SCS than MY (0.153 vs. 0.251, respectively).

### 3.3. Genetic and Environmental Correlations

Correlations between teat scoring and production traits for all random effects are reported in Table 3. Correlations were considered significant when the 95% empirical confidence intervals did not include the ‘0’ value. 

Genetic correlations for TS and TS_a_ with MY were positive and significant, showing values of 0.862 and 0.893, respectively. The correlation between MY and TS_b_ was not significant. Only the genetic correlation between TS and SCS was significant showing a value of 0.439.

None of the cow permanent environmental correlations of teat scoring traits with MY were significant. All correlations with SCS were significant and positive but weak, showing values of 0.141, 0.125 and 0.152 for TS, TS_a_ and TS_b_. 

For the herd environmental correlations, the only significant estimate was found between TS_b_ and MY with a value of 0.463. However, for all estimates the tendency was for a positive relationship between teat scoring traits and MY and a negative relationship with SCS.

### 3.4. Influence of Fixed Effects on Teat Scores

Least square mean estimates for the hygiene score are reported in Table 4 while estimates for the teat position effect are reported in Table 5. Estimates are expressed on the phenotypic scale (probability of a TS greater than 1 for TS_a_ or TS greater than 2 for TS_b_).

The hygiene effect did not appear to be relevant for any traits, since the posterior distributions of the estimates overlapped.

A relationship was observed among teat positions and TS_a_. Indeed front quarters showed a higher probability of a TS greater than 1 or TS greater than 2. The same pattern was found for TS_b_, with probabilities for front left and front right quarters at 0.095 and probabilities for rear left and rear right quarters at 0.094. While the front-back contrast could not be declared significant at the chosen α threshold of 0.05, the trend appeared evident.

Least square mean estimates expressed on the phenotypic scale (i.e., probability) for each parity by month of lactation effect are in Figure 3. Results for TS_a_ are in the upper part while results for TS_b_ in the lower part of the figure.

Estimates for TS_a_ in lactation 1 showed an increase in the higher scores towards the end of lactation with estimates going from below 0.25 to 0.30. Values similar to the end of lactation 1 were maintained in lactation 2 and subsequently, suggesting that the deterioration of teats is carried over (it is worth considering that this is a cross-sectional study, and not a longitudinal study). Estimates within lactations 2 and subsequently did not show as clear a pattern, though a small increase was shown until the peak of lactation. Lactation estimates across stages for TS_a_ increased from 0.28 to 0.47 to 0.51 in lactation 1, 2 and 3, respectively. Similarly, lactation estimates across stages for TS_b_ increased from 0.094 to 0.097 to 0.106 for lactation 1, 2 and subsequently. Conversely, estimates within-lactation for TS_b_ did not show relevant differences.

### 3.5. Genetic Response in Teat Score Traits

The genetic response in teat score traits (TS, TS_a_ and TS_b_) after selecting for both MY and SCS applying different relative emphasis is reported in Figure 4. The x-axis gives, from left to right, the relative emphasis as shifting from MY to SCS, the y-axis gives the genetic response in genetic standard deviation units for the respective trait given the overall relative emphasis. The values on the left side of the graph show the response when low emphasis is given to MY, which is positively (unfavorably) correlated with the three traits, and high emphasis is given to SCS, which is still positively correlated but it was given negative emphasis. 

When the emphasis on MY was below 12 (and subsequently the emphasis on SCS was over −88) genetic response for all three teat score traits was negative, i.e., teat score improved. The response for TS became positive at the value of MY emphasis of 17 (i.e., −83 for SCS), became positive for TS_a_ at the MY emphasis value of 14 (i.e., −86 for SCS) and at the value of 22 for TS_b_ (i.e., −78 for SCS). All the response curves reached a plateau around the value of 40 for emphasis on MY. At large values of MY emphasis, genetic response became strong and positive (unfavorable) for TS (0.56 genetic standard deviation units), moderately positive for TS_a_ (0.32 genetic standard deviation units) and almost null for TS_b_ (0.09 units). The genetic response was therefore mostly unfavorable for all traits unless most of the emphasis was given to SCS (at least 90%). While genetic correlations with other productive and reproductive traits are missing, we could infer that strong index emphasis should be given to udder health if the deterioration of teat score at the genetic level needs to be avoided. The response was stronger, in either direction, for TS than TS_a_, probably due to the larger heritability since the genetic correlations with MY and SCS were about the same for both traits. The response was small to null for TS_b_.

## 4. Discussion

The results of the present study confirm that THK has a genetic component. Indeed, heritability ranged from 0.084 for TS_b_ (binary trait) to 0.238 for TS (scored using a 1 to 4 scale). These results agree well with the estimates previously reported by Chrystal et al. [17] but are lower than those presented by Lojda et al. [37], Seykora and McDaniel [38], and Chrystal et al. [16]. An interesting aspect regards the scale used to score THK which might influence heritability estimates. Pantoja et al [11], in their systematic review of the association between THK and mastitis, support the idea that THK should be scored using a reduced number of categories. This approach will guarantee an adequate number of records per THK score, hence a better statistical power. However reducing the number of THK scores will partly jeopardize its actual variation and, as [17] suggested, possibly lead to lower heritability estimates. In the present study, the relative frequency of a TS = 4 (i.e., very rough callous ring) was below 2%, lower than that reported by Pantoja et al [11]. However, the 95% highest probability density intervals of the posterior mean heritability estimate for TS trait ranged from 0.116 to 0.364, suggesting that a 4-point scale is still a viable solution. A similar pattern was observed for cow repeatability, with larger estimates when THK was scored on a 4-point scale. This result is quite interesting because it suggests that reducing the scale and treating the THK as a binary trait could be a less accurate measure if recorded only once per cow. 

Parity and DIM strongly affect THK [11] and the results found in the present study, even if modulated by the scoring scale, confirm this finding. 

The relationship between THK and other breeding goals (e.g., milk yields and somatic cell score) has been mainly investigated considering the former as the dependent variable (Y) and the latter as possible factors affecting Y. The relationship, especially with SCS, was not always clear and significant [11,16,17]. In the present study a different approach has been applied. Indeed, previous studies, not only in dairy cattle [3] but also in other dairy species [39], have suggested a non-zero genetic correlation between udder traits and milk and/or SCS. This aspect is not negligible and should be taken into consideration when implementing a selection index. If the genetic correlation is statistically different from zero there could have been a correlated favorable or unfavorable response on THK. This is particularly true if we consider that selection on milk yield has been the primary objective in dairy cattle for many years [40]. The genetic correlation between THK and milk yield was large and positive, i.e., unfavorable, confirming previous findings [41] in dairy cattle. However, a recent and large study by Tribout [42] in dairy cattle did not confirm the existence of a single quantitative trait locus (QTL) with pleiotropic effect on both milk production and udder morphology suggesting instead the presence of neighboring QTLs that show linkage disequilibrium, eventually leading to a non-null genetic correlation. When using THK scored on 1–4 scale, the genetic correlation was positive and significant (95% empirical confidence intervals not including the ‘0’ value). A positive genetic correlation means that an increase in SCS causes an increase in THK hence selection for lower **SCS** should have a positive impact on teat-end condition. When the relative emphasis on milk yield is below 15%, the response in teat score traits is favorable. Nowadays the focus of selection has moved away from being purely production oriented toward a more balanced breeding goal, even if milk yield has still a relative emphasis not less than 40–50% [40]. 

In this study we also estimated cow-level correlations as well as herd-test-day correlations. While the former expresses trait associations that are driven by any cow-related factor other than additive genetics, the latter expresses an association at the herd-level, which is likely due to management alone. In management, we include any strategic choice made by the farmer as well as the equipment used for milking, which is known to affect the insurgence of THK. 

Cow correlations were not significant when MY was the correlated trait, but were positive, though weak, when SCS was the correlated trait. This suggests that, beyond genetics, there is a common factor that drives SCS to increase and THK to appear more frequently. At the herd level, correlations were not significant when TS and TS_a_ were the correlated traits. TS_b_ was positively correlated with MY and negatively correlated with SCS. This suggest that herd management (and equipment) that leads to higher MY also leads to higher THK. Also, the herd management that leads to lower SCS also leads to higher THK (moving from classes 1 and 2 to classes 3 and 4, specifically). 

The different direction of the correlation between TS_b_ and SCS depending on the additive genetic, cow permanent or herd level suggests that there are counter-acting factors. While cows themselves could show a positive association for which higher SCS means higher THK, the herd management (and equipment) shows the opposite direction, i.e., herds with lower SCS have higher THK and vice versa. The milking practice, for example, could be a factor determining this association. 

## 5. Conclusions

In the current study, multivariate analysis conducted on 10,776 scoring records and 30,160 production records from 2469 Italian Holstein cows enabled the estimation of the genetic parameters for teat-end score and its relationship with milk yields and SCS. Teat-end score has a genetic background and is genetically related to both production and SCS. The scale used to classify teat-end score has an impact on the magnitude of the estimates which however were always statistically different from zero. Teat-end score was also genetically related to SCS, reinforcing the idea that udder morphology is still a fundamental piece in the control of mammary infection. Finally, an unfavorable genetic correlation of teat score with milk yield was observed. However, considering that the current selection objectives implemented in dairy cattle worldwide have shifted toward a more balanced breeding goal a further genetic deterioration in teat-score is not expected. The importance of the negative effects of some environmental aspects, such as milking routine, should not be forgotten.

## Figures and Tables

**Figure 1 animals-10-02271-f001:**
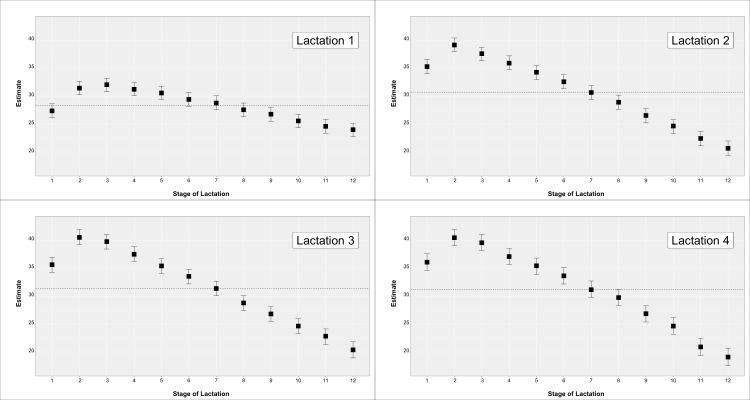
Milk yield across parity and stage of lactation for cows with a teat-score evaluation.

**Figure 2 animals-10-02271-f002:**
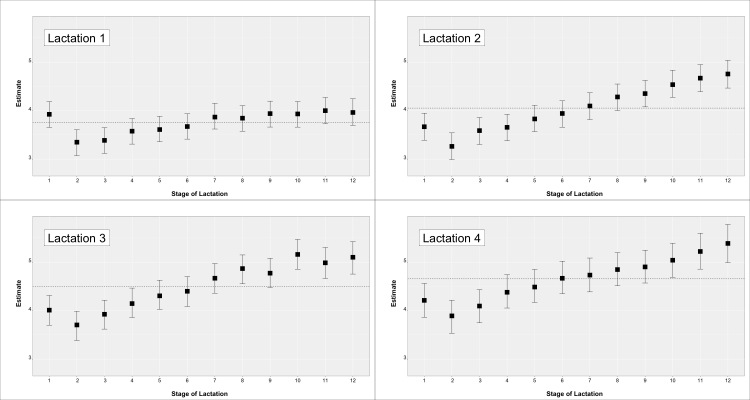
Somatic cell score across parity and stage of lactation for cows with a teat-score evaluation.

**Figure 3 animals-10-02271-f003:**
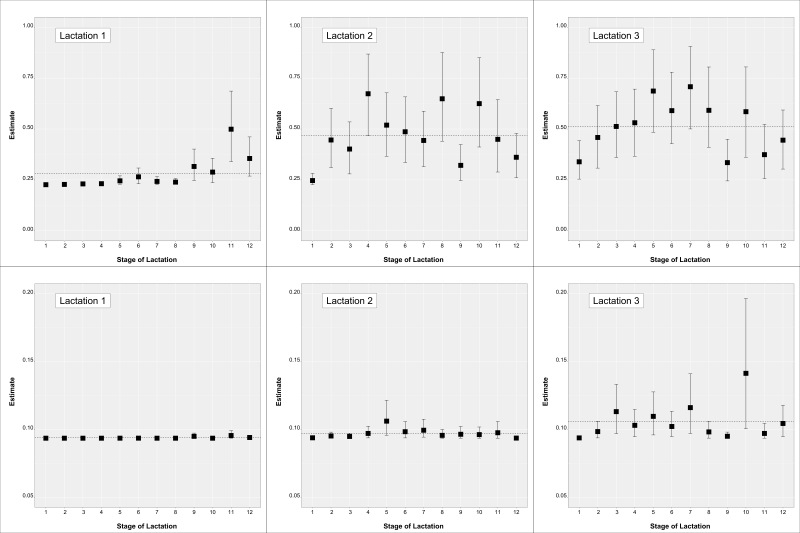
Least square mean estimates expressed on the phenotypic scale (posterior mean and 95% highest probability density intervals) for the lactation by stage of lactation effect on TS_a_ (upper figures) and TS_b_ (lower figures) traits. The dashed horizontal line represents the estimate for that lactation across stages.

**Figure 4 animals-10-02271-f004:**
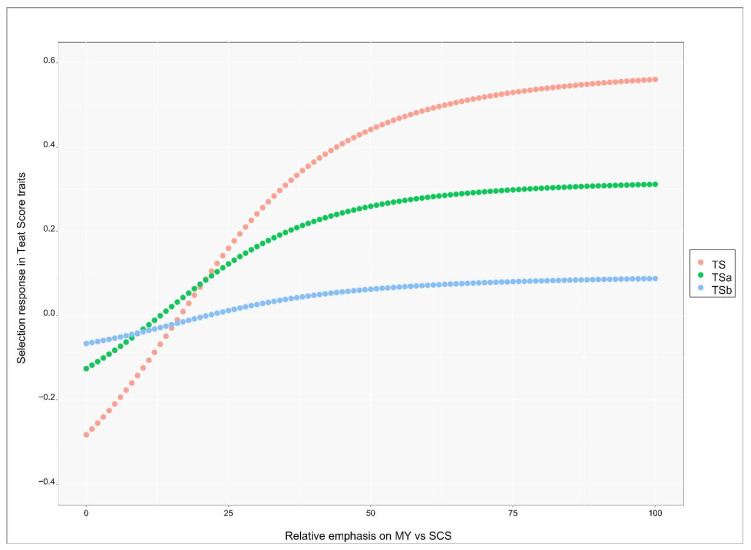
Expected genetic progress in Teat Score traits (y-axis) when shifting relative economic emphasis (x-axis) on milk yield (MY) vs. somatic cell score (SCS). Negative values mean an improvement in teat score (less teat-end hyperkeratosis).

**Table 1 animals-10-02271-t001:** Frequency of observations for each teat score in the dataset (*n* = 10,776).

Score	Number of Records	Relative Frequency		
		TS ^1^	TS_a_ ^1^	TS_b_ ^1^
1	7078	65.7	65.7	90.6
2	2687	24.9	34.3	
3	838	7.78		9.4
4	173	1.61		

^1^ TS = teat scoring trait based on a four-classes ordinal scoring (1 = absent callosity, 2 = smooth callous ring, 3 = rough callous ring, 4 = very rough callous ring). TS_a_: binary teat scoring trait where classes 2, 3 and 4 were combined into a single class. TS_b_: binary teat scoring trait where classes 1 and 2 vs. 3 and 4 were combined.

**Table 2 animals-10-02271-t002:** Estimates (posterior mean and 95% highest probability density intervals) of variance components and genetic parameter estimates for the traits considered in this study.

Parameter	Trait				
	TS	TS_a_ ^1^	TS_b_ ^1^	MY ^1^	SCS ^1^
Sire genetic variance	0.438	0.128	0.036	0.958	0.148
	(0.233; 0.702)	(0.072; 0.188)	(0.010; 0.066)	(0.450; 1.608)	(0.084; 0.214)
Cow permanent environmental variance	3.998	1.449	0.430	15.12	1.488
	(3.465; 4.532)	(1.324; 1.577)	(0.371; 0.488)	(14.05; 16.15)	(1.398; 1.591)
Herd-test day variance	1.648	0.506	0.160	15.04	0.624
	(0.902; 2.548)	(0.288; 0.768)	(0.078; 0.249)	(7.85; 23.06)	(0.292; 0.925)
Residual variance	1.281	1.073	1.085	28.01	1.788
	(1.217; 1.343)	(1.051; 1.095)	(1.060; 1.105)	(27.57; 28.51)	(1.758; 1.817)
Heritability (h^2^)	0.238	0.162	0.084	0.065	0.146
	(0.116; 0.364)	(0.090; 0.235)	(0.025; 0.155)	(0.030; 0.109)	(0.080; 0.210)
Intra-Herd heritability (h^2^-IH)	0.289	0.248	0.250	0.126	0.234
	(0.153; 0.45)	(0.061; 0.434)	(0.135; 0.363)	(0.025; 0.217)	(0.114; 0.343)
Cow repeatability	0.544	0.460	0.251	0.257	0.368
	(0.469; 0.603)	(0.417; 0.503)	(0.222; 0.282)	(0.220; 0.290)	(0.333; 0.401)
Herd-test-day repeatability	0.221	0.159	0.093	0.251	0.153
	(0.141; 0.313)	(0.096; 0.224)	(0.051; 0.141)	(0.164; 0.353)	(0.093; 0.224)

^1^ TS = teat scoring trait based on a four-classes ordinal scoring (1 = absent callosity, 2 = smooth callous ring, 3 = rough callous ring, 4 = very rough callous ring). TS_a_: binary teat scoring trait where classes 2, 3 and 4 were combined into a single class. TS_b_: binary teat scoring trait where classes 1 and 2 vs. 3 and 4 were combined. MY = daily milk yield. SCS = daily somatic cell score.

**Table 3 animals-10-02271-t003:** Estimates (posterior mean and 95% highest probability density intervals) of genetic correlations between teat score and production traits.

Item	MY ^1^			SCS ^1^		
	TS ^1^	TS_a_ ^1^	TS_b_ ^1^	TS ^1^	TS_a_ ^1^	TS_b_ ^1^
Genetic correlation	0.86	0.89	0.54	0.44	0.36	0.38
	(0.71; 0.98)	(0.61; 0.99)	(−0.11; 0.96)	(0.10; 0.80)	(−0.02; 0.70)	(−0.09; 0.90)
Cow permanent environmental correlation	0.01	0.02	−0.003	0.14	0.12	0.15
	(−0.04; 0.06)	(−0.03; 0.07)	(−0.08; 0.06)	(0.09; 0.19)	(0.07; 0.18)	(0.08; 0.22)
Herd-test day Correlation	0.30	0.30	0.46	−0.25	−0.25	−0.47
	(−0.06; 0.64)	(−0.03; 0.62)	(0.13; 0.72)	(−0.57; 0.10)	(−0.58; 0.08)	(−0.76; −0.17)

^1^ TS = teat scoring trait based on a four-classes ordinal scoring (1 = absent callosity, 2 = smooth callous ring, 3 rough callous ring, 4 = very rough callous ring). TS_a_: binary teat scoring trait where classes 2, 3 and 4 were combined into a single class. TS_b_: binary teat scoring trait where classes 1 and 2 vs. 3 and 4 were combined. MY = daily milk yield. SCS = daily somatic cell score.

**Table 4 animals-10-02271-t004:** Least square mean estimates (posterior mean and 95% highest probability density intervals) for the influence of Hygiene score on TS_a_ and TS_b_ traits. Estimates are expressed on the phenotypic scale (probability of a TS greater than 1 for TS_a_ or TS greater than 2 for TS_b_).

Trait	Hygiene Score ^2^			
	1	2	3	4
TS_a_ ^1^	0.326	0.364	0.382	0.317
	(0.285; 0.375)	(0.314; 0.423)	(0.309; 0.451)	(0.248; 0.397)
TS_b_ ^1^	0.093	0.094	0.094	0.095
	(0.092; 0.094)	(0.094; 0.094)	(0.094; 0.096)	(0.094; 0.097)

^1^ TS_a_: binary teat scoring trait where TS classes 2, 3 and 4 were combined into a single class. TS_b_: binary teat scoring trait where TS classes 1 and 2 vs. 3 and 4 were combined. ^2^ Hygiene of udder, flanks and legs was scored based on a 4-point scale system, from very clean (score 1) to very dirty skin (score 4).

**Table 5 animals-10-02271-t005:** Least square mean estimates (posterior mean and 95% highest probability density intervals) for the influence of udder quarter on on TS_a_ and TS_b_ traits. Estimates are expressed on the phenotypic scale (probability of a TS greater than 1 for TS_a_ or TS greater than 2 for TS_b_).

Trait	Udder Quarters			
	Front Left	Front Right	Rear Left	Rear Right
TS_a_ ^1^	0.393	0.388	0.302	0.312
	(0.327; 0.462)	(0.323; 0.455)	(0.265; 0.34)	(0.271; 0.356)
TS_b_ ^1^	0.095	0.095	0.094	0.094
	(0.093; 0.097)	(0.093; 0.097)	(0.092; 0.096)	(0.092; 0.096)

^1^ TS_a_: binary teat scoring trait where TS classes 2, 3 and 4 were combined into a single class. TS_b_: binary teat scoring trait where TS classes 1 and 2 vs. 3 and 4 were combined.

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
