# Peer review of "Heritability of Teat Condition in Italian Holstein Friesian and Its Relationship with Milk Production and Somatic Cell Score"

_animals, 2020, doi:10.3390/ani10122271_

Round 1

Reviewer 1 Report

The paper is covering a well-known risk factors for mastitis. The aim of the paper if really ambitious, but the scientific approach is very disappointing. Scientific literature, as reported in the paper, suggests that teat shape have also a role, and we can deduct that genetic could have an influence through “modelling” teat shape. However, the outcome of THK even if we assume a genetic predisposition is, indeed, a pathological outcome which has milking machine and milking procedures as  main risk factors. If the risk factors are not applied very likely THK does not appear. A cow could present THK, when risk factors are present, but the same cow in the same herd once these risk factors are removed, in many cases and in 2-3 month will be THK disappears. None of these aspects have been covered in the paper, and presumptively considered for the analysis of data.

Other comments

Lines 44-57 There are several papers covering these aspects in the last 40 years. This part of the introduction covers these aspects with few lines and with references that has relatively poor relationship with the text (ref 1-7) as well as ref 12 which is not the most relevant for the statement, while the hyperkeratosis aspects are summarized in 3 lines (49-51).

No information are supplied on herd characteristics, milking machine and milking procedures applied in the herds, cow characteristics (parity, days in milk) all factors that affect the outcome of THK. All these aspects seem to be solved in the analysis as cow characteristics and environmental effects, and confirmed in discussion, but this is not sufficient to evaluate the quality of the data used to assess genetic effects.

No information are supplied on how peoples scoring teats have been instructed and certified to avoid bias due to individual scorer. Otherwise nearly 90 lines were devoted to explain statistical analysis. Moreover, the combination of THK score in two different way is arbitrary and  debatable.

Lines 117-122, 191-195 the term “incidence” cannot be used in this context since there is no evaluation of time before the outcome. The same problem is repeated several times in the other part of the paper.

Discussion is relatively short, and some of the statements are quite controversial (lines 358-359) “A positive genetic correlation means that an increase in SCS causes an increase in THK hence  selection for less SCS should have a positive impact on teat-end condition”. I’m really curious how the Authors will be able to explain this relationship from biological point of view, since the actual evidence, supported by a very large number of papers, is the opposite. Moreover, as Authors knows very well, SCS is also affected by mastitis outcome and mastitis is correlated to THK, but also to other factors (herd hygiene, milking hygiene, prevalence of bacteria species…) and none of these aspects were considered seeking a genetic correlation between SCS and THK.

Author Response

REV 1

The paper is covering a well-known risk factors for mastitis. The aim of the paper if really ambitious, but the scientific approach is very disappointing. Scientific literature, as reported in the paper, suggests that teat shape have also a role, and we can deduct that genetic could have an influence through “modelling” teat shape. However, the outcome of THK even if we assume a genetic predisposition is, indeed, a pathological outcome which has milking machine and milking procedures as  main risk factors. If the risk factors are not applied very likely THK does not appear. A cow could present THK, when risk factors are present, but the same cow in the same herd once these risk factors are removed, in many cases and in 2-3 month will be THK disappears. None of these aspects have been covered in the paper, and presumptively considered for the analysis of data.

AU: Undoubtedly, our study presents some limitations. As or the disappearance of THK, our manuscript leans on a cross-sectional study which does not allow to follow the THK across time. However, several lactations and stages of lactations are represented. While a longitudinal study would be more appropriate to understand the dynamics behind THK disappearance, a multi-factorial cross-sectional experimental design should be sufficient to estimate genetic parameters.

It would have been very useful to have records of teat shape in this study. That could have helped us dissect the phenotypic correlation into its genetic and environmental components. We could also have studied genetic variation of THK marginally on teat shape by fitting its effect in the statistical model.

However, once again, for the study of THK conditionally on all its components we believe that our dataset was sufficient to achieve the goal. Results suggest a positive role of selection for reduced SCS in driving down the genetic risk for THK.

Other comments

Lines 44-57 There are several papers covering these aspects in the last 40 years. This part of the introduction covers these aspects with few lines and with references that has relatively poor relationship with the text (ref 1-7) as well as ref 12 which is not the most relevant for the statement, while the hyperkeratosis aspects are summarized in 3 lines (49-51).

AU: We appreciate this suggestion. Unfortunately, information on milking equipment and practices was not available or this study. We still believe though, that we were able to absorb the impact of those actors with the herd effect.

No information are supplied on herd characteristics, milking machine and milking procedures applied in the herds, cow characteristics (parity, days in milk) all factors that affect the outcome of THK. All these aspects seem to be solved in the analysis as cow characteristics and environmental effects, and confirmed in discussion, but this is not sufficient to evaluate the quality of the data used to assess genetic effects.

AU: Information on the number of herds and their location have been added.

No information are supplied on how peoples scoring teats have been instructed and certified to avoid bias due to individual scorer. Otherwise nearly 90 lines were devoted to explain statistical analysis. Moreover, the combination of THK score in two different way is arbitrary and  debatable.

AU: Teat scoring was in charge of a small (n=4) group of technicians who have been efficiently trained. Information have been added in the manuscript.

Lines 117-122, 191-195 the term “incidence” cannot be used in this context since there is no evaluation of time before the outcome. The same problem is repeated several times in the other part of the paper.

AU: We appreciate this suggestion. The term ‘incidence’ has been replaced with ‘relative frequency’ throughout the manuscript.

Discussion is relatively short, and some of the statements are quite controversial (lines 358-359) “A positive genetic correlation means that an increase in SCS causes an increase in THK hence  selection for less SCS should have a positive impact on teat-end condition”. I’m really curious how the Authors will be able to explain this relationship from biological point of view, since the actual evidence, supported by a very large number of papers, is the opposite. Moreover, as Authors knows very well, SCS is also affected by mastitis outcome and mastitis is correlated to THK, but also to other factors (herd hygiene, milking hygiene, prevalence of bacteria species…) and none of these aspects were considered seeking a genetic correlation between SCS and THK.

AU: We appreciate this comment.

We believe that selection for lower SCS would lead to lower THK at the genetic level. The intent of our study was not to find the biological explanation to the phenomenon but simply to describe it in a parametric way and provide an overlook of the potential deterioration of teat health due to the use of some breeding goals. Other researchers, perhaps in veterinary science, could help address this question.

We want to emphasize the results on the correlations, not only at the genetic level. Cow permanent environmental correlations are positive as genetic correlations, although weaker in magnitude. This suggest that cow-inherent actors other than additive genetic background lead to a positive association between THK and SCS.

Herd-level correlations are, instead, negative though only TSb shows a significant estimate. This suggests that herds with higher SCS show lower TSb and vice versa. Herd level correlations should be exclusively due to herd management, since the single-cow effects are removed from the herd component. While the models used do not allow to establish causation between SCS and THK, we are still safe to assume that some herd practices may drive down SCS but increase THK or increase SCS while maintaining low insurgence of THK. Hence, they might mitigate, if not cancel, the positive association between SCS and THK. This could lead to phenotypic correlations that are not equal (in sign and magnitude) to genetic correlations.

A section has been added to the Discussion

Reviewer 2 Report

The reliability of genomic prediction is influenced by several factors, including the size of the reference population, which makes genomic prediction for breeds with a relatively small population size challenging, such as Italian Holstein Friesian dairy cattle. The study of the genetic component using statistical methods and the elucidation of its relationship to milk yield and somatic cell count is of great importance for the future health of herds.

I have a few questions that will help future authors who want to research other breeds of cows or other animal species. Questions:

In section "2.1. Dataset", please complete the information: Which regions in Italy did 48 cow herds come from? what age range were the cows (or average age)? in what time period (years) the research was conducted?

I believe that the authors should add the information in the discussion of the results that, according to many studies, the age of the cows, the number of lactations, etc., have a significant influence on MY and the SCS.

I wish you health. Let the cowid virus never be in your family.

Author Response

REV 2

The reliability of genomic prediction is influenced by several factors, including the size of the reference population, which makes genomic prediction for breeds with a relatively small population size challenging, such as Italian Holstein Friesian dairy cattle. The study of the genetic component using statistical methods and the elucidation of its relationship to milk yield and somatic cell count is of great importance for the future health of herds.

I have a few questions that will help future authors who want to research other breeds of cows or other animal species. Questions:

In section "2.1. Dataset", please complete the information: Which regions in Italy did 48 cow herds come from? what age range were the cows (or average age)? in what time period (years) the research was conducted?

AU: Thanks for the suggestion. We have added the information you asked for

I believe that the authors should add the information in the discussion of the results that, according to many studies, the age of the cows, the number of lactations, etc., have a significant influence on MY and the SCS.

AU: We agree with reviewer 2 that age and parity have a large effect both on milk and scs. Indeed there are a lot of works about it. However even if our objective was mainly to focus on the relationship among these traits and THK we have added 2 new figures with the phenotypic MILK and SCS trend across parity and stage of lactation

I wish you health. Let the cowid virus never be in your family.

AU: Best wishes to you and your family!

Reviewer 3 Report

Title: The article regarding the Heritability of teat condition in Italian Holstein Friesian and its relationship with milk production and somatic cell score

Francesco Tiezzi, Antonio Marco Maisano, Stefania Chessa, Mario Luini and Stefano Biffani

Reviewer:

This article is about the heritability of teat condition in Italian Holstein cow related to milk production, somatic cell score, and infection. This is an important area of research addressing the quality of milk production issues with possible somatic cell scores. The authors have articulated the research finding appropriately that would be of greater interest to the reader. However, this reviewer urges for minor revision addressing each comment given below and recommend for publication.

This reviewer has the following comments:

  1. Line 61 and 62 author says teat diameter but no units. Please provide the units.
  2. Line 67 authors have abbreviated SCS but failed to mention the full version of the word or phrase. I guess SSC means somatic cell score.
  3. Is somatic cell count (SCC, line 66-67) is same as SCS? If so the author should consider one terminology that would be less confusing to the general reader.

Author Response

REV 3

This article is about the heritability of teat condition in Italian Holstein cow related to milk production, somatic cell score, and infection. This is an important area of research addressing the quality of milk production issues with possible somatic cell scores. The authors have articulated the research finding appropriately that would be of greater interest to the reader. However, this reviewer urges for minor revision addressing each comment given below and recommend for publication.

This reviewer has the following comments:

  1. Line 61 and 62 author says teat diameter but no units. Please provide the units.

AU: THK is generally scored using an ordinal scale. This definition has been added to the text (line 66)

  1. Line 67 authors have abbreviated SCS but failed to mention the full version of the word or phrase. I guess SSC means somatic cell score.

AU: Changed. Now lines 77-78.

  1. Is somatic cell count (SCC, line 66-67) is same as SCS? If so the author should consider one terminology that would be less confusing to the general reader.

AU: Thanks for the comment, we fixed that. Now line 67.

Reviewer 4 Report

This is a study aiming at investigating association between teat end condition and milk yield and udder health. It also describes the role of breeding selection on teat end condition. The study is well-written and the topic and research question are very interesting and inspiring. The text is concise (in a good way).

However, I think material and methods should be explained in a bit more detail for some areas (see comments below). Some of the results are not easy to understand, because heritability is a very specific area of research including specific vocabulary. Discussion is also interesting and some results are very well discussed, but others are not discussed at all (see below).  

Specific comments

Simple Summary:

line 20: milk somatic cell count

Abstract:

line 26: dairy cattle production

line 34 and throughout the study: milk yields and somatic cell score… singular and plural words are mixed sometimes, especially considering these words.

line 35-36: This sentence is a conclusion, and it would help the reader to differentiate it from the results if you rephrased it a bit, for example: When the emphasis of breeding is set towards milk yield…

line 37-39: This is quite heavily said, based on your results. In your text (line 363) you say that selection is still for 40-50% based on milk yield. Your data shows that it should be less than 10% if you want better teat end condition, doesn`t it?

Keywords: teat-end hyperkeratosis with lower case

Introduction:

line 44: one of the most

line 54: change should be reworded. For example alteration would sound better. However, the whole sentence leads the reader to wonder what other alterations or deteriorations in teat condition might there be?  

line 58: as a consequence

line 62: repeatability of what? Measurement of teat diameter and shape?

line 64: heritability of what? Teat diameter?

line 67: after being…to facilitate analysis to somatic cell score (SCS)…

The objectives should be in past tense

Material and Methods

2.1. You should start this section with the information of the herds, animals etc.

line 90, reference is missing for hygiene scoring

line 100: Please rephrase “lactation of teat scoring”

line 101: if there are around 10 000 teat scoring records, it means that these 2600 cows were recorded once. How come there are 30 000 production records? Please explain this in full detail. Who did teat end scoring? Why were milk samples taken? Was scoring made before or after milking? etc…

line 105: What does this mean: teat scoring and production measures were considered as repeated and independent… How can they be both?

2.2.

line 186: something is missing before ..is the g1 is the first column of G…

Results

You should not repeat the values of the tables in the text, for example lines 191-194 and elsewhere.

In the Tables you should use different superscripts instead of just number 1.

Line 222: surprisingly is a word used in discussion, not in results.

Table 2 and 3: I do not quite get what you mean by environmental correlation or Herd test-day correlation. Could you please explain? In the text you speak about herd environmental correlation (line 244)

Overall I have great trouble to understand what is meant by the estimates in Table 4 and 5. Is it the estimate probability of getting a TSa score of 2? Could this be written open more clearly in the legend of the Table?

Figure 1: This is very clear and informative figure, thank you! In the legend you should explain TSa and b. Stage of lactation is months, and should be mentioned. And again, least square estimates of what? Propability of…? Lines 277-278 is repetition of the legend.

Line 288: and subsequently

Line 291: Do you mean lactation 3 instead of stage 3?

Figure 2: You have nicely explained this figure for the reader, thank you.

Line 311: should it be over 88

Line 313: should be (i.e. 83 for SCS)

Line 319 to 324 belongs to discussion

Line 345-346: Isn`t this vice versa?

Line 349: This aspect is not negligible.

Line 355: QTL? Please explain.

Discussion is a bit short. I missed discussion of lactation stage and parity (more than one sentence), hygiene, teat position, environmental correlation (?), repeatabilities.

Author Response

REV 4

This is a study aiming at investigating association between teat end condition and milk yield and udder health. It also describes the role of breeding selection on teat end condition. The study is well-written and the topic and research question are very interesting and inspiring. The text is concise (in a good way).

However, I think material and methods should be explained in a bit more detail for some areas (see comments below). Some of the results are not easy to understand, because heritability is a very specific area of research including specific vocabulary. Discussion is also interesting and some results are very well discussed, but others are not discussed at all (see below).  

AU: Thanks for this suggestion. Two more paragraphs have been added to the end of the Discussion section. We would appreciate some further feedback from the Reviewer.

Specific comments

Simple Summary:

line 20: milk somatic cell count

AU: Fixed, thank you.

Abstract:

line 26: dairy cattle production

AU: We added ‘industry’, thank you.

line 34 and throughout the study: milk yields and somatic cell score… singular and plural words are mixed sometimes, especially considering these words.

AU: We fixed this throughout the study, thank you.

line 35-36: This sentence is a conclusion, and it would help the reader to differentiate it from the results if you rephrased it a bit, for example: When the emphasis of breeding is set towards milk yield…

AU: We appreciate this suggestion. We have reworded this sentence.

line 37-39: This is quite heavily said, based on your results. In your text (line 363) you say that selection is still for 40-50% based on milk yield. Your data shows that it should be less than 10% if you want better teat end condition, doesn`t it?

AU: Thanks for this suggestion. In reality we need about less than 15-20% of emphasis to be placed on MY to avoid a deterioration, better if less than 50%. The sentence at the end of the abstract was rephrased.

Keywords: teat-end hyperkeratosis with lower case

AU: Fixed, thank you.

Introduction:

line 44: one of the most

AU: Fixed, thank you.

line 54: change should be reworded. For example alteration would sound better. However, the whole sentence leads the reader to wonder what other alterations or deteriorations in teat condition might there be?  

AU: line 68

line 58: as consequence

AU: Fixed, thank you.

line 62: repeatability of what? Measurement of teat diameter and shape?

AU: Correct. The sentence was rephrased.

line 64: heritability of what? Teat diameter?

AU: Correct, but across lactations. This information was added.

line 67: after being…to facilitate analysis to somatic cell score (SCS)…

AU: Fixed, thank you.

The objectives should be in past tense

AU: Fixed, thank you.

Material and Methods

2.1. You should start this section with the information of the herds, animals etc.

AU: Information have been added

line 90, reference is missing for hygiene scoring

AU: Reference has been added

line 100: Please rephrase “lactation of teat scoring”

AU: Rephrased, now lines 122-124.

line 101: if there are around 10 000 teat scoring records, it means that these 2600 cows were recorded once. How come there are 30 000 production records? Please explain this in full detail.

AU: In this study, we were not able to match the scoring date with a production record since they came from different data collection efforts. The records were not aligned, but several test-day records were available for each cow. A sentence was added, now line XX.

Who did teat end scoring? Why were milk samples taken? Was scoring made before or after milking? etc…

AU: These information have been added to text. A group of 4 trained technicians was in charge of teat scoring. Teat scoring was usually made before milking. The recording of teat score was part of a larger project which included, among the others, milk sampling for bacteriological analyses and SCC.

line 105: What does this mean: teat scoring and production measures were considered as repeated and independent… How can they be both?

AU: Correct. They came from two different datasets and were not aligned. We have tried to exploit all available information. We had one information for teat scoring but we have information about lactation test-day. We think that they are pretty informative

2.2.

line 186: something is missing before ..is the g1 is the first column of G…

AU: Only the first column of G is used in the cross-product with w.

Results

You should not repeat the values of the tables in the text, for example lines 191-194 and elsewhere.

AU: Thanks for the suggestion. However we think that it could be useful for the reader to repeat some values. Doing so he will not need to look always at the table. We prefer to leave the way it is and let the editor decide

In the Tables you should use different superscripts instead of just number 1.

AU: We are not sure about it. We think that in each table the numbering is independent and not sequential. However we are going to follow editor’s suggestions

Line 222: surprisingly is a word used in discussion, not in results.

AU: Right. We removed it.

Table 2 and 3: I do not quite get what you mean by environmental correlation or Herd test-day correlation. Could you please explain? In the text you speak about herd environmental correlation (line 244)

AU: We appreciate this suggestion. A whole paragraph was added to the end of the discussion section.

Overall I have great trouble to understand what is meant by the estimates in Table 4 and 5. Is it the estimate probability of getting a TSa score of 2? Could this be written open more clearly in the legend of the Table?

AU: We have added the following text to Table legend: “Estimates are expressed on the phenotypic scale (probability of a TS greater than 1 for TSa or TS greater than 2 for TSb)”.

Figure 1: This is very clear and informative figure, thank you! In the legend you should explain

TSa and b. Stage of lactation is months, and should be mentioned. And again, least square estimates of what? Propability of…? Lines 277-278 is repetition of the legend.

AU: We have added the missing information

Line 288: and subsequently

AU: corrected

Line 291: Do you mean lactation 3 instead of stage 3?

AU: We actually referred to the peak of second lactation, which fell in stage 3. However that was confusing and therefore removed.

Figure 2: You have nicely explained this figure for the reader, thank you.

AU: Thank you!

Line 311: should it be over 88

AU: Fixed, thank you!

Line 313: should be (i.e. 83 for SCS)

AU: SCS actually got a negative emphasis.

Line 319 to 324 belongs to discussion

AU: Thanks for this suggestion. We find that some explanatory sentence should appear in the Results section as well, to help the reader.

Line 345-346: Isn`t this vice versa?

AU: The relationship between SCS and THK is positive but actually unfavorable because it means a worst teat condition as long as SCS increase. That’s because we would need to select for less SCS, it will eventually have a positive impact on THK as well.

Line 349: This aspect is not negligible.

AU: Fixed, thank you.

Line 355: QTL? Please explain.

AU: The sentence was partially rephrased, thank you.

Discussion is a bit short. I missed discussion of lactation stage and parity (more than one sentence), hygiene, teat position, environmental correlation (?), repeatabilities.

AU: Thanks for this suggestion. We have added a section on the cow and herd correlations. Regarding the stage and parity, we don’t feel that we should argument further given the cross-sectional nature of our study. A longitudinal study would be preferable to assess the change in TS across lactations and their stages.

Round 2

Reviewer 1 Report

The new version of the paper is not addressing properly the critical issues raised in the previous revision, therefore my opinion is unchanged.

Author Response

We deeply regret that rev 1 didn't appreciate our work. We do agree that THK is  largely affected by an environmental component however not all individuals (i.e. cows) have the same response. Indeed, there is a genetic component, even if THK can change across/within lactation.

Reviewer 4 Report

This version has improved, thank you very much. However I still have some minor comments, many of them considering the same matters than last time because I did not always get explanations I think were necessary. Material and methods are better now. The biggest problem in this text is that the results are not easy to understand, referring to a very special subject. However, the writers have explained most of it in the discussion, and that is very helpful.

Line 59: you tell that genetic component of teat condition has been studied before, but the reference 16 is about teat end shape, which is a different thing (and again on line 72-76). I think you show no references about THK and its genetics?  

Line 64: heritability of what? Teat diameter? This was not corrected in this second version.

Line 67: somatic cell score (SCS) using a logarithmic transformation…

Line 90: Thank you for more information about the materials and methods. However, you told that teat scoring was usually made before milking. If it was sometimes done right after milking, I mean right after the teat cups had been taken off, the score could be somewhat different than when analyzing the teat score before attaching the teat cups.

So how was it: teat cleaning - teat scoring – milk sampling – milking; a systematic order of investigation or not? This could have a great effect on the results. One can easily score for example teat score 3 before milking as 4 after milking, this I have seen in practise.

Line 90: You told that the recording of teat score was part of a larger project which included, among the others, milk sampling for bacteriological analyses and SCC. You should tell this to a reader and include a reference if you already have.

Line 97: I suppose you should use different numbering for the references, and not put number 41 (last reference) in between the former numbers..” References must be numbered in order of appearance in the text”

Line 113-114: Last time I asked: line 105: What does this mean: teat scoring and production measures were considered as repeated and independent… How can they be both?

You answered: Correct. They came from two different datasets and were not aligned. We have tried to exploit all available information. We had one information for teat scoring but we have information about lactation test-day. We think that they are pretty informative.

So, you must mean that you considered the data as repeated because there were many milk yields for one cow, that I can understand. But what was independent? Please explain in the text. And how did you get 40936 records? This has to be clear for the reader.

Line 195: I still don`t get this, this sentence must not be right? G is the variance-covariance between the teat score trait, MY and SCS; is the g1 is the first column of G… There are 2 verbs is and no subject in this sentence.

Line 372: can you please explain what does QTL stand for?

Author Response

We would like to thank rev 4 for his/her useful suggestions. 

Please find our answers to your last comments:

Line 59: you tell that genetic component of teat condition has been studied before, but the reference 16 is about teat end shape, which is a different thing (and again on line 72-76). I think you show no references about THK and its genetics?  

AU This is a good point. Indeed, trait definition might not be exactly the same and this is the reason why we are talking about “the genetic component of teat condition”. However the term THK is nowadays the most common and officially used to identify the physiological process of adaptation of the mammary gland. As a matter of fact the methodology used in the present study (Neijenhuis et al. [2000]) depends on teat-end shape and callosity.

Line 64: heritability of what? Teat diameter? This was not corrected in this second version.

AU fixed

Line 67: somatic cell score (SCS) using a logarithmic transformation…

AU fixed

Line 90: Thank you for more information about the materials and methods. However, you told that teat scoring was usually made before milking. If it was sometimes done right after milking, I mean right after the teat cups had been taken off, the score could be somewhat different than when analyzing the teat score before attaching the teat cups.

So how was it: teat cleaning - teat scoring – milk sampling – milking; a systematic order of investigation or not? This could have a great effect on the results. One can easily score for example teat score 3 before milking as 4 after milking, this I have seen in practise.

AU this is another valuable observation. Unfortunately it is not possible to have a definite answer because we don’t have such a piece of information. However we supposed that it could be absorbed by the herd-visit effect. It’s reasonable to hypothesize that even if the scoring was made after/before milking it was consistent within herd-visit

Line 90: You told that the recording of teat score was part of a larger project which included, among the others, milk sampling for bacteriological analyses and SCC. You should tell this to a reader and include a reference if you already have.

AU the original project is included in the acknowledgments, however we added a reference to another paper from the same project.

Line 97: I suppose you should use different numbering for the references, and not put number 41 (last reference) in between the former numbers..” References must be numbered in order of appearance in the text”

AU References numbering has been fixed

Line 113-114: Last time I asked: line 105: What does this mean: teat scoring and production measures were considered as repeated and independent… How can they be both?

You answered: Correct. They came from two different datasets and were not aligned. We have tried to exploit all available information. We had one information for teat scoring but we have information about lactation test-day. We think that they are pretty informative.

So, you must mean that you considered the data as repeated because there were many milk yields for one cow, that I can understand. But what was independent? Please explain in the text. And how did you get 40936 records? This has to be clear for the reader.

AU we have slightly modified the original text  and we have added an additional explanation (line 106-130)

Line 195: I still don`t get this, this sentence must not be right? G is the variance-covariance between the teat score trait, MY and SCS; is the g1 is the first column of G… There are 2 verbs is and no subject in this sentence.

AU fixed

Line 372: can you please explain what does QTL stand for?

AU added quantitative trait loci